# Characterisation of the First Bovine Parainfluenza Virus 3 Isolate Detected in Cattle in Turkey

**DOI:** 10.3390/vetsci6020056

**Published:** 2019-06-13

**Authors:** Harun Albayrak, Zafer Yazici, Emre Ozan, Cuneyt Tamer, Ahmed Abd El Wahed, Stefanie Wehner, Kristina Ulrich, Manfred Weidmann

**Affiliations:** 1Department of Virology, Faculty of Veterinary Medicine, Ondokuz Mayis University, Samsun 55200, Turkey; harunalbayrak55@msn.com (H.A.); zyazici@omu.edu.tr (Z.Y.); cuneyt_tamer@hotmail.com (C.T.); 2Department of Laboratory Animals, Faculty of Veterinary Medicine, Ondokuz Mayis University, Samsun 55200, Turkey; ozanemre1983@hotmail.com; 3Division of Microbiology and Animal Hygiene, University of Goettingen, 37077 Goettingen, Germany; abdelwahed@gwdg.de; 4Institute of Aquaculture, University of Stirling, Scotland FK9 4LA, United Kingdom; wehner.ste@gmail.com (S.W.); ulrich_kristina@yahoo.com (K.U)

**Keywords:** Bovine parainfluenza virus type 3, Turkey

## Abstract

A respiratory disease outbreak on a cattle farm in northern Turkey produced respiratory tract symptoms and severe pneumonia symptoms in 20 calves. Eight calves died, and a lung specimen from one carcass was analysed for bacteria and for viruses of the Bovine respiratory diseases complex. Bacteriological analysis was negative, but antigen detection ELISA and RT-PCR results indicated the presence of Bovine parainfluenza virus (BPIV). Virus isolation succeeded on Madin-Darby Bovine Kidney cells, and subsequent whole genome sequencing and phylogenetic analysis identified BPIV-3c. This is the first report of BPIV-3c isolation from cattle in Turkey, indicating the need for more virological and epidemiological studies.

## 1. Introduction

Viruses and bacteria—in combination with stress factors—have an important role in bovine respiratory disease, and the induced febrile diseases are commonly known as “Shipping Fever” [1]. The Bovine respiratory diseases complex (BRDC) is a major health problem for cattle worldwide [2,3]. It consists of *bovine parainfluenza virus* type 3 (BPIV-3), *bovine herpes virus type-1* (BHV-1), *bovine respiratory syncytial virus* (BRSV) and *bovine viral diarrhea virus* (BVDV), which all lead to severe respiratory disease in cattle [1].

BPIV-3 is an enveloped, nonsegmented, negative-sense and single-stranded RNA virus belonging to the *Respirovirus* genus in the family *Paramyxoviridae* [4,5]. The genome of BPIV-3 consist of approximately 15,500 bases encoding six large open reading frames (ORFs) that encode six proteins, including the nucleocapsid protein (N), phosphoprotein (P), matrix protein (M), hemagglutinin/neuraminidase (HN), fusion protein (F) and large protein (L) [6].

To date, based on phylogenetic analysis, three known BPIV-3 genotypes, designated A(BPIV-3a), B(BPIV-3b) and C(BPIV-3c), have been described [2,6]. Genotype A was first isolated in the United States, but has also been isolated in Egypt, China and Japan [6,7]. Genotype B was first reported in Australia [4]. Isolations of genotype C were conducted in China, South Korea and Japan [2,3,8]. Furthermore, all three genotypes were reported in Argentina [9].

BPIV-3 has been serologically detected in several domestic and free-ranging ungulates, including cattle, goats, sheep, camels and new-world camelids [5]. Cross-species infections have been reported in numerous instances, including BPIV-3 in sheep [10] and humans [11]. In cattle, the clinical manifestations of BPIV-3 range from subclinical to acute respiratory disease with a variety of symptoms, including high fever, nasal discharge and coughing. Cattle infected with BPIV-3 are generally co-infected with other viral and bacterial pathogens (*Mycoplasma bovis, Pasteurella multocida, Mannheimia haemolytica,* etc.). This indicates that BPIV-3 may be immunosuppressive and pave the way for secondary infections [5]. In spite of serological evidence in cattle and small ruminants, to date there have been no reported BPIV-3 isolations from cattle in Turkey. Currently, the SF-4 strain of BPIV-3 (genotype A) provided from abroad is used for serological studies [12,13].

In this study, we report (i) the first isolation of BPIV-3 from cattle in Turkey and (ii) present the results of sequencing and phylogenetic analysis of the complete BPIV-3 genome.

## 2. Materials and Methods

### 2.1. Sample Material

In May 2016, a local respiratory disease outbreak was reported to have occurred at an animal breeding farm that had 65 Holstein-Friesian cattle (20 calves and 45 cows) in Ordu Province (41°03′ N, 37°49′ E) in northern Turkey. The carcasses of seven dead calves were destroyed, and one carcass was sent to the Samsun Veterinary Control Institute for analysis. The animal protocols used in this work were evaluated and approved by the Samsun Veterinary Control Institute Ethics Committee (Protocol: 19572899-P.900/34). They are in accordance with Animal welfare guidelines and the National law for Laboratory Animal Experimentation (Law no. 5199/28914).

### 2.2. Preparation of Lung Homogenate

For cell culture isolation, approximately 1000 mg of lung tissue was homogenised in Minimal Essential Medium (MEM) (5 mL/1 g tissue) containing 2% penicillin/streptomycin (Sigma, UK) for 5 min using a Silent Crusher M tissue homogeniser (Heildolph, Germany) at 4 °C, and centrifuged at 3000 rpm for 10 min. The supernatant was filtered through 0.22 µm filters (Millipor, USA) and stored at –20 °C until used.

For RT-PCR, approximately 30 mg of lung tissue was homogenised by using a Tissue-Lyser (Qiagen AG, Switzerland) in 1.8 mL MEM containing 2% penicillin/streptomycin (Sigma, UK). Obtained homogenates were clarified by centrifuging at 9000× rpm for 15 min. The supernatant was stored at –20 °C until used.

### 2.3. Viral and Bacterial Culture

Madin-Darby Bovine Kidney (MDBK) cells were used for virus isolation. Cells and all chemicals (pestiviruses-free as defined by the manufacturer) that were used for this study were checked by RT-PCR [14] and were all found to be pestivirus-free. Briefly, lung homogenate was inoculated in MDBK cells. The inoculated cells were incubated at 37 °C, 5% CO_2_ for 1 h before adding Dulbecco’s modified essential medium (DMEM, Gibco, UK) containing 5% fetal serum (Gibco, UK) and 1% antibiotic solution (Sigma-Aldrich, UK). The cells were checked daily under an inverted microscope (Olympus, Japan) and monitored for the appearance of cytopathic effects (CPEs). When the CPEs were visible in 80% of cells, the cell culture was harvested, disrupted by three freeze–thaw cycles and centrifuged at 1500 rpm for 10 min in order to remove cell debris. The virus containing supernatant was stored at –20 °C.

The homogenates were also inoculated onto blood agar (supplemented with 7% sheep blood) and MacConkey agar and incubated aerobically at 37 °C for 1–2 days (24–48 h).

### 2.4. Antigen ELISA

In order to investigate respiratory viruses, including BVDV, BHV-1, BPIV-3 and BRSV, lung homogenate from dead animals was tested using the Pulmotest Antigen Elisa Kit (Bio X diagnostic, Rochefort, Belgium, Cat No: Bio K 340/2) according to the manufacturer’s instructions. Interpretation of results was performed after the plates were read using a 450 nm filter, and ODs were then calculated using the following formula: Value = delta OD sample × 100/delta OD positive.

### 2.5. RT-PCR Detection

Total RNA was extracted from all samples (lung homogenate and cell culture supernatants) using the RNeasy Mini Kit (Qiagen, Hilden, Germany) according to the manufacturer’s directions. RT-PCR was performed using specific primers (BPI fw 5′-CATTGAATTCATACTCAGCAC-3′ and BPI rev 5′-AGATTGTCGCATTT(AG)CCTC3-’) to amplify the expected size of 400 bp for the Fusion (F) protein gene sequence of BPIV-3 [15]. The RT-PCR mixture was prepared in a total volume of 50 µL containing 5 µL of sampleRNA, 10 µL of 5x mastermix, 1 µL of dNTP mix, 2 µL of each primer, 2.5 µL of DTT, 1 µL of enzyme mix and 26.5 µL of water. Amplification was conducted using the following thermal profile: a room temperature (RT) step at 50 °C for 30 min, initial denaturation at 95 °C for 15 min, 30 cycles consisting of a denaturation step at 94 °C for 45 s, an annealing step at 51 °C for 45 s, an elongation step at 72 °C for 1 min and a final extension cycle at 72 °C for 10 min. At the end of amplifications, 10 µL of each PCR product were loaded on a 1.5% agarose gel stained with ethidium bromide and were visualized by using Quantum gel imaging and documentation system (Vilber, Frankfurt, Germany) after running on 100 V for 40 min.

### 2.6. Sequencing and Phylogenetic Analysis

Ten microlitres (10 µL) of extracted RNA were converted into ss-cDNA using the Super Script III RT Kit (Life Technologies, UK) and ds-cDNA was synthesized using the NEBNext^®^ mRNA Second Strand Synthesis Module (New England Biolabs, Hitchin, UK) by adding 20 µL of ss-cDNA to 60 µL of the ds-cDNA master mix and following the manufacturer’s instructions.

Ds-cDNA was cleaned by adding 80 µL of Ampure XP beads (Beckman Coulter, High Wycombe, UK) per sample (1x) and incubation for 5 min at room temperature. The sample was placed into a magnet rack so the clear supernatant could be discarded. The magnetic beads pellet was washed twice with 200 µL of freshly prepared 70% Ethanol, while keeping the tube in the magnet, and dried in the magnet for 5 min at room temperature to remove all ethanol traces. The pellet was eluted in 25 µL of molecular-grade water and transferred into a new Eppendorf DNA LoBind 1.5 mL tube. The DNA was stored at –20 °C until used for library preparation via an Illumina Nextera XT DNA Library Preparation Kit (Illumina, Cambridge, UK).

Raw reads were filtered to remove low-quality and low-complexity sequences using PRINSEQ (v0.20.4). Filtered sequences were down-sampled using KHMER (v2.0). Remaining sequences were assembled by spades (v3.7.1) [16,17,18]. Sequences were aligned using neighbour joining in the ClustalW multiple alignment option in the MEGALIGN Pro module of DNASTAR 10.0.1 using 1000 bootstrap replicates. The phylogenetic tree was edited in DENDROSCOPE 3.2. [19].

## 3. Results

At the farm, all cattle presented with respiratory tract symptoms, e.g., high fever, coughing, nasal/ocular discharge and dyspnea. Furthermore, severe pneumonia symptoms were evident among all 20 calves. In spite of antibiotic intervention (daily dosage: penicillin 16,000 IU/kg, streptomycine 20 mg/kg) to control suspected bacterial infection (*Mycoplasma bovis, Pasteurella multocida, Mannheimia haemolytica,* etc.), eight calves died.

### 3.1. Laboratory Results

A lung specimen was analysed for bacteria and for viruses of the BRDC complex. Bacterial growth was not detected on any of the inoculated bacterial culture plates. Homogenized lung samples tested for respiratory viral pathogens scored positive for BPIV-3 and negative for BoHV-1, BRSV and BPIV-3 antigens in the antigen ELISA. All positive and negative controls scored correctly. The OD values of the samples were determined to be higher than those of a positive control (136%). A CPE was observed 96 h after inoculation onto MDBK cells (Figure 1) and sustained over four passages in order to increase the titre. RNA extracts of the supernatant and cells of each passage were tested by BPIV-3 RT-PCR. The predicted specific fragment for the fusion protein gene of approximately 400 bp was amplified in all of these extracts.

### 3.2. Sequencing and Phylogenetic Analysis

A BPIV-3 RT-PCR positive sample of passage four was sequenced using Illumina chemistry and the MiSeq platform. Sequencing resulted in 1,860,581 raw reads. After filtering, 1,033,799 reads remained. The mean coverage of the 15,504-nt-long scaffold was 4161.37 nt. Phylogenetic analysis clearly placed the sequence of the Turkish BPIV-3 isolate in the genotype C group (Figure 2).

## 4. Discussion

BPIV-3 is now recognized to be one of the most important viral respiratory pathogens of cattle, has also been associated with BRDC development in feedlot cattle [20,21,22], and continues to cause serious economic losses for the global cattle industry. BVDV and BHV-1 infections are very common in Turkey [13]. Other important viruses, such as BPIV-3 and BRSV, have not yet been isolated in Turkey prior to this study, and their roles in BRDC are still obscure in Turkey.

Genome sequencing and phylogenetic analysis of the first isolate of BPIV-3 from a lung sample of a deceased calf in Turkey groups the isolate into genotype BPIV-3c, which so far has only been described to be in circulation in parts of Asia and South America. The isolation of BPIV-3c from a diseased calve calls for a wider epidemiological study to determine if more isolates can be identified and if BPIV-3c or even other BPIV genotypes are circulating in Turkey. A very recent paper indeed reports RT-PCR detection, and the determination of a partial sequence, of BPIV-3c from nasal swabs of Holstein cattle in the Erzurum-Aşkale Province in Turkey [23].

Turkey imports livestock (especially cattle) from more than 15 countries from three continents, including Europe, South-America and Australia, but there is no livestock and animal product trade between Turkey and South Asian countries, in particular China. It is, therefore, unclear if the BPIV-3c isolate from Turkey is part of a Eurasian subclade due to the proximity of its sequence to those of Chinese isolates (Figure 2) or if BPIV-3c might have entered Turkey by way of cattle imports from South-America.

It is not clear if the readouts obtained using the BPIV SF4 strain used in neutralisation assays for serological surveillance in Turkey determine protection against possibly circulating BPIV-3c strains.

Vaccines used against genotype B and C, although available on the international market, are currently not licensed for use in Turkey. A recent analysis of commercial BPIV-3a containing vaccines administered in the United States indicated that these induce low neutralisation titres, especially against genotype B and C [1,6]. Thus, even if these vaccines were used in Turkey, they might not be effective to control possibly circulating BPIV-3c.

This investigation into a case of respiratory disease in calves in the Ordu Province identified the first BPIV-3c isolate from Turkey; however, the overall epidemiology of BPIV-3 in Turkey needs to be investigated to be able to assess if vaccines should be introduced.

## Figures and Tables

**Figure 1 vetsci-06-00056-f001:**
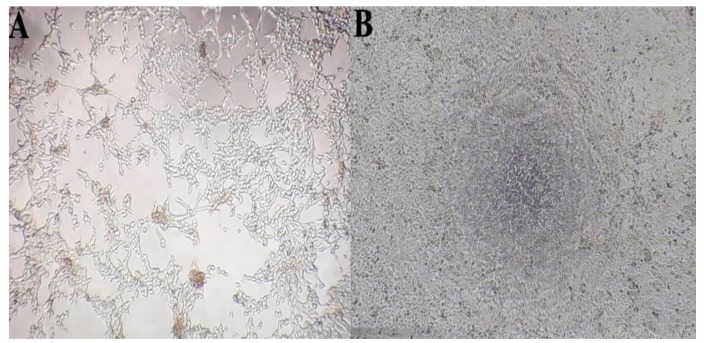
Cytopatheic effect induced by the Bovine parainfluenza virus (BPIV)_3c isolate on MDBK cells 300 dpi. (Magnification: 400x). **A**: Infected cells; large areas of the monolayer have lysed. **B**: negative control.

**Figure 2 vetsci-06-00056-f002:**
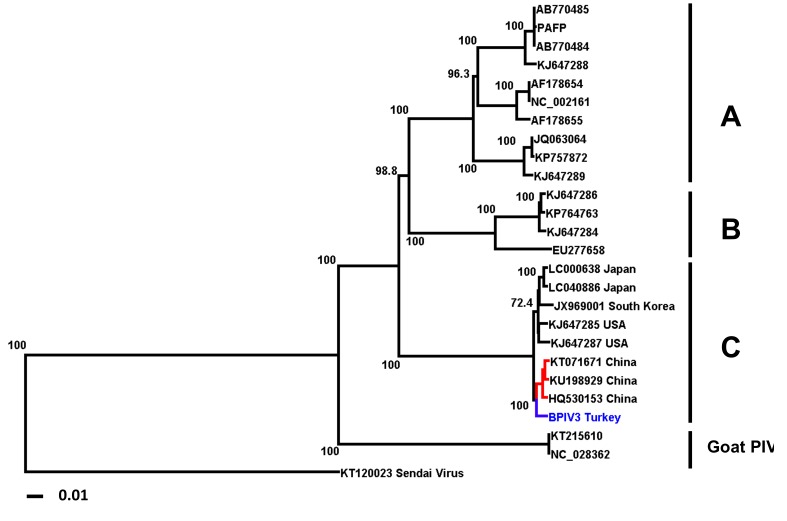
Phylogenetic tree of published full genome sequences of BPIV-3 strains. Strains are identified by their accession numbers. High bootstrap values are indicated. The Turkish isolate is highlighted in blue (Genebank accession no. MH357343). Chinese isolates are highlighted in red.

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
