# Peer review of "Characterisation of the First Bovine Parainfluenza Virus 3 Isolate Detected in Cattle in Turkey"

_vetsci, 2019, doi:10.3390/vetsci6020056_

Round 1
Reviewer 1 Report
General comments
This is a very interesting and well written paper concerning the first report of BPIV-3c isolation from cattle in Turkey. Considering the importance of Bovine Respiratory Disease Complex as a major health problem for cattle worldwide, this case report conveys important information on the topic, which is very relevant to the journal’s aims and thus it is suggested to be accepted for publication in “Veterinary Sciences”.
Specific comments
There are a couple of minor comments which might improve the manuscript.
· Line 21: It should be better to use “bacteriological” than “microbiological”. Microbiological analysis is the use of laboratory methods for the detection and/or identification of microorganisms such as bacteria, fungi, viruses, parasites, so it is better to use the first term, since ELISA and RT-PCR were both positive for BPIV.
Line 37: Authors should use “six” instead of “6”.
Line 49: “Fever” is also known as “pyrexia”, unless they mean “high fever and fever”.
Line 50, “co-infected with other viral and bacterial pathogens”: In my opinion this information might be included in the text, providing specific examples.
Line 81: Why MacConkey agar was incubated for 1 up to 7 days? Which members of Enterobacteriaceae could be isolated after 7 days of incubation? Authors should be more specific.
Lines 121-129: “In May 2016 ….. BRDC complex”. Authors should simply state the findings of their study in the “Results” section. This paragraph contains information relating to “Materials and Methods” section.
Lines 169-170: Authors should rewrite to clarify.
Author Response
Response to Review
Please find bellow our detailed responses to the comments of the reviewers.
Reviewer 1
Specific comments
There are a couple of minor comments which might improve the manuscript.
Line 21: It should be better to use “bacteriological” than “microbiological”. Microbiological analysis is the use of laboratory methods for the detection and/or identification of microorganisms such as bacteria, fungi, viruses, parasites, so it is better to use the first term, since ELISA and RT-PCR were both positive for BPIV.
Done
Line 37: Authors should use “six” instead of “6”.
Done
Line 49: “Fever” is also known as “pyrexia”, unless they mean “high fever and fever”.
Pyrexia was deleted.
Line 50, “co-infected with other viral and bacterial pathogens”: In my opinion this information might be included in the text, providing specific examples.
The following bracket listing examples was added (Mycoplasma bovis, Pasteurella multocida, Mannheimia haemolytica, etc.)
Line 81: Why MacConkey agar was incubated for 1 up to 7 days? Which members of Enterobacteriaceae could be isolated after 7 days of incubation? Authors should be more specific.
The sentence was adapted to:
The homogenates were also inoculated onto blood agar (supplemented with 7% sheep blood) and MacConkey agar and incubated aerobically at 37°C for 1-2 days (24-48hours).
Lines 121-129: “In May 2016 ….. BRDC complex”. Authors should simply state the findings of their study in the “Results” section. This paragraph contains information relating to “Materials and Methods” section.
Part of the paragraph was moved into the Methods section and information on ethical approval was added.:
2.1. Sample material
In May 2016, a local respiratory disease outbreak was reported from an animal breeding farm that had 65 Holstein-Friesian cattle (20 calves and 45 cows) in Ordu Province (41°03′ N, 37°49′ E), in Northern Turkey. Carcasses of seven dead calves were destroyed while one carcass was sent to the Samsun Veterinary Control Institute for analysis. The animal protocols used in this work were evaluated and approved by the Samsun Veterinary Control Institute Ethics Committee (Protocol: 19572899-P.900/34). They are in accordance with Animal welfare guidelines and the National law for Laboratory Animal Experimentation (Law no. 5199/28914).
The Results pargraph has been edited to:
At the farm all cattle presented with respiratory tract symptoms e.g. high fever, coughing, nasal/ocular discharge, dyspnea. Futhermore severe pneumonia symptoms were evident among all 20 calves. In spite of antibiotic intervention (daily dosage: penicillin-16000 IU/kg, streptomycine-20mg/kg) to control suspected bacterial infection (Mycoplasma bovis, Pasteurella multocida, Mannheimia haemolytica, etc.) eight calves died.
Lines 169-170: Authors should rewrite to clarify.
The paragraph was edited:
Vaccines used against genotype B and C although available on the international market, are currently not licenced for use in Turkey. A recent analysis of commercial BPIV-3a containing vaccines administered in the USA indicated that these induce low neutralisation titres, especially against genotype B and C [1,6]. Thus even if these vaccines were used in Turkey they might not be effective to control possibly circulating BPIV-3c.
Reviewer 2 Report
Albayrak et al., present a case report on an important bovine respiratory virus in Turkey. They claim that this is the first report, however Timurkan et al., (J Vet Res/63/2019) have already presented report on BPIV3 and BRSV from Turkey in cattle. The authors have neither cited nor acknowledged this study. Hence, it cannot be claimed as the first report from Turkey.
Specific Comments:
1. Keywords are essentially BPIV3 written twice.
2. Line 51-53 is inaccurate. Please refer to citation above.
3. Sample Material: What is the rational behind choosing only one animal? How did the authors choose to work on that particular animal?
4. Lines 104-107. Use consistent acronym for double stranded DNA.
Author Response
Response to Review
Please find bellow our detailed responses to the comments of the reviewers.
Reviewer 2
Albayrak et al., present a case report on an important bovine respiratory virus in Turkey. They claim that this is the first report, however Timurkan et al., (J Vet Res/63/2019) have already presented report on BPIV3 and BRSV from Turkey in cattle. The authors have neither cited nor acknowledged this study. Hence, it cannot be claimed as the first report from Turkey.
and
Line 51-53 is inaccurate.
We thank the reviewer for pointing out this very recent report of which we were indeed not aware and have now acknowledged by including the following sentence into the discussion. However this paper describes PCR detection and partial sequencing but not isolation of BPIV3-3c, therefore we have not changed our statement that we report the first isolation of BPIV-3c from Turkey.
A very recent paper indeed reports RT-PCR detection and the determination of a partial sequence of BPIV-3c from nasal swabs of Holstein cattle in the Erzurum-Aşkale Province in Turkey [23].
23. Timurkan OM, A.H., Sait A. Identification and molecular characterisation of bovine parainfluenza virus-3 and bovine respiratory syncytial virus: first report from Turkey. Journal of Veterinary Research 2019, 63, doi:10.2478/jvetres-2019-0022.
Specific Comments:
1. Keywords are essentially BPIV3 written twice.
The abbreviation was deleted
2. Sample Material: What is the rationale behind choosing only one animal? How did the authors choose to work on that particular animal?
The responsible field veterinarian decided to send only one calf clear disease symptoms for laboratory diagnosis.
4. Lines 104-107. Use consistent acronym for double stranded DNA.
Done
We have also updated figure 1 now also showing a negative control.
Updated figure legend:
Figure 1. Cytopatheic effect induced by the BPIV_3c isolate on MDBK cells 300 dpi. (Maginfication: 400x). A: Infected cells, large areas of the monolayer have lysed. B: negative control.
Round 2
Reviewer 2 Report
The authors have addressed my previous concerns in their updated report and response.